# TransAbs: Taming Absolute Interaction for Efficient Relative Motion Prediction

## Abstract

Accurate motion prediction in complex social scenarios requires capturing effective spatial interactions among traffic participants. Limited by the unsatisfactory accuracy of absolute interaction, recent methods transform the absolute scenario into multiple relative scenarios to obtain high-quality predictions. However, this relative prediction suffers from re-encoding relative motion information (*e.g.*, spatial position and participants interaction), leading to computational inefficiency compared to absolute interaction. In this paper, we present TransAbs, which tames the absolute interaction to achieve relative effects, while maintaining the computational efficiency. The core idea of TransAbs is to map the Hadamard representation between a pair of absolute positions to their relative position. Incorporating the absolute positional embedding to the attention formulation (Hadamard product first and then sum up), TransAbs achieves relative positional embedding and spatial interaction simultaneously by leveraging a post-multiplication positional encoding. To align the Hadamard presentation and attention scores, TransAbs is optimized jointly with the motion predictor. We evaluate the effectiveness of TransAbs by integrating it into the transformer-based motion predictor commonly employed in motion prediction. Extensive experiments on a large scale public benchmark, Waymo Open Motion Dataset, demonstrate that TransAbs successfully balances prediction accuracy and computational efficiency with minimal overhead—achieving comparable accuracy while eliminating the redundant re-encoding introduced by relative interaction.

## 1 Introduction

Motion prediction is essential for understanding traffic participants behavior and serves as a bridge between perception and decision-making in autonomous driving Hu et al. (2023). In recent years, significant progress has been made in this field Shi et al. (2021; 2023); Gu et al. (2021); Shi et al. (2022); Liu et al. (2024). However, a key challenge remains: effectively capturing spatial interactions among traffic participants in highly dynamic social environments, driving the need for more efficient interaction strategies.

To model spatial interaction, two mainstream strategies are widely used in motion prediction. The first is absolute interaction Ngiam et al. (2021), where all agents in a scenario are represented in an absolute coordinate system originated by scenario center or self-driving car as shown in Figure 1. (2). This strategy employs a shared coordinate system to predict future trajectories for all agents with one-time forward pass. However, the absolute coordinate system suffers from high variance coordinate values and the model is dependent on the specific coordinate system of the scene and struggles to learn generalizable motion patterns.

To improve prediction accuracy, most methods employ a relative interaction Shi et al. (2022; 2024); Zhang et al. (2024) to model spatial interaction. Concretely, they build relative trajectories or positions with any pair of traffic participants based on the rigid transformation (translation and rotation) of motion trajectory as illustrated in Figure 1. (3). This approach generally achieves higher prediction accuracy compared to absolute strategy. However, it requires re-encoding each participants individually, leading to computational inefficiency.

To address this issue, we first analyze the process of relative interaction. For a target participant as illustrated orange vehicle in Figure 1. (3), we translate and then rotate all participants to normalize

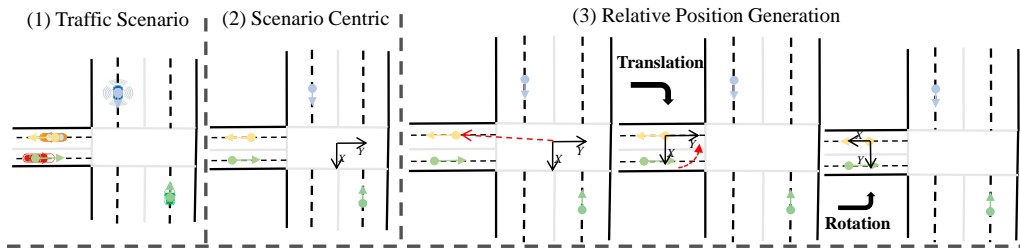

Figure 1: Illustration of spatial interaction. (1) is a traffic scenario with 4 agents and multiple map vectors. (2) is the absolute interaction centered on the traffic scenario center. (3) is an example of relative interaction centered on the orange vehicle. This process is repeated for all participants (agents and map vectors), leading to expensive computation. Best view in color.

the coordinate system aligning with the target participant. Based on this normalized coordinate system, the self-attention mechanism is used to model spatial interaction. The whole relative interaction is finished by repeating this process for all agents and map vectors. Through this analysis, the re-encoding stems from the relative position generation centered on each target participant. However, we identify that the relative position generation is similar with self-attention if we regard the target participant as query and other participants as the key. The difference lies in the attention calculation, where the self-attention uses the vector inner product to interact with keys, while the relative position generation uses two transformations to interact with keys. This identification motivates us to ask: *Can we incorporate relative position generation into the self-attention to eliminate the redundant re-encoding process?*

To answer this question, we propose TransAbs, which involves a novel auto-transformer module to integrate relative position generation into the self-attention process, thus achieving efficient spatial interaction. To achieve that, the auto-transformer transforms the Hadamard representation of a pair of absolute positions into their relative position by a self-supervised learning. The popularly used stacked transformer encoders incorporate the transformed absolute position into the attention calculation (Hadamard dot first and then sum up) by a post-multiplication manner to achieve relative position encoding and spatial interaction simultaneously, thus eliminating the re-encoding process caused by relative interaction.

To align the Hadamard presentation and attention scores, TransAbs is optimized jointly with the motion predictor. We evaluate the effectiveness of TransAbs by integrating it into the current state-of-the-art transformer based motion prediction methods. Experimental results demonstrate that TransAbs successfully balances prediction accuracy and computational efficiency with minimal overhead—achieving comparable accuracy while eliminating the redundant re-encoding introduced by relative interaction. The contributions of this paper are summarized as follows: 1) We analyze the process of relative interaction modeling and identify the re-encoding stems from the relative position generation centered on each target participan. We propose TransAbs, which tames the absolute interaction to achieve relative effects, while maintaining the computational efficiency; 2) We design our TransAbs as an auto-transformer architecture, which transforms the Hadamard representation between a pair of absolute positions into their relative position. It is incorporated into the self-attention formulation with a post-multiplication positional encoding to model spatial position and interaction simultaneously; 3) We evaluate our TransAbs by integrating it into the commonly used transformer-based motion predictor. TransAbs successfully balances prediction accuracy and computational efficiency with minimal overhead—achieving comparable accuracy while eliminating the redundant re-encoding.

## 2 RELATED WORKS

### 2.0.1 ABSOLUTE INTERACTION

To model spatial interactions, a naive approach directly utilizes the captured spatial positions of each participant as input features. This strategy employs a shared coordinate system centered on

a reference point (*e.g.*, scenario center or ego vehicle) to predict future trajectories for all agents through a single forward pass Casas et al. (2020); Ngiam et al. (2021). In this interaction strategy, displacement vectors are commonly employed to reduce coordinate value variance, while absolute position embeddings are used to recover spatial information Mangalam et al. (2020). However, absolute positions inherently suffer from high coordinate variance, rendering the model dependent on the specific coordinate of each scene and consequently hindering the learning of generalizable motion patterns.

### 2.0.2 RELATIVE INTERACTION

To improve prediction accuracy, most methods transform absolute interactions into multiple relative interactions by leveraging rigid transformations of motion trajectories and map polylines. The fundamental operation involves constructing target-centric coordinate systems centered on the current position and direction of specific target participants.

Two primary approaches exist for relative interactions: full relative interaction and present-only relative interaction. Full relative interaction Gao et al. (2020); Liu et al. (2021a); Shi et al. (2022); Zhou et al. (2022); Gao et al. (2023); Liu et al. (2024); Kang et al. (2025); Yan et al. (2025) transforms the observed states of all participants into target-centric coordinate systems. For scenarios with $N$ target participants, this approach necessitates constructing $N$ target-centric coordinate systems and executing the entire model $N$ times, resulting in substantial computational overhead.

To mitigate computational costs, present-only relative interaction Shi et al. (2024); Zhang et al. (2024); Zhou et al. (2023); Jia et al. (2023); Cui et al. (2022) decouples target and surrounding participants. Specifically, this method extracts target motion states from each self-centric coordinate system without considering surrounding participants, thereby obtaining temporal information for each target participant through a single forward pass. To preserve spatial information, the method transforms the current absolute position into relative positions. A relative positional encoder similar to relative position embedding Shaw et al. (2018) then encodes these relative positions and directions to provide spatial information for the interaction module, *i.e.*, the attention mechanism. In addition, there have another approach, rotary positional embedding Su et al. (2024), is used to encode relative position, while it cannot represent the periodicity of agent heading Zhao et al. (2025).

Both relative interaction approaches suffer from the computational inefficiency of individually re-encoding each participant's entire or current motion states. In contrast, our TransAbs avoids the re-encoding to achieve relative effects while maintaining computational efficiency.

## 3 METHOD

We present TransAbs, a novel approach that achieves relative spatial interaction effects through absolute position encoding without requiring computationally expensive position re-encoding. As illustrated in Figure 2, our framework consists of three main components: (1) a motion tokenizer that generates agent tokens, map tokens, and absolute position embeddings; (2) a stacked transformer encoder enhanced with our TransAbs module for efficient relative spatial modeling; and (3) a stacked motion decoder for trajectory prediction.

### 3.1 PROBLEM DEFINITION

In a traffic scenario involving $N$ agents (vehicles, pedestrians, and bicycles) and $M$ scene elements (map polylines), the motion predictor observes the historical states of all entities from time 0 to $t$, and predicts the future trajectories of the $N$ agents from time $t + 1$ to $T$. To accurately represent the positional relationships among all traffic participants, both agents and scene information are unified within an absolute coordinate system. The $N$ agents are represented as $\mathbf{A} \in \mathbb{R}^{N \times t \times c_a}$, where $c_a$ denotes the feature dimension (including position, heading, velocity, etc.) of an agent at each time step. The $M$ map polylines are denoted as $\mathbf{M} \in \mathbb{R}^{M \times l \times c_m}$, where $l$ is the number of points in a map polyline and $c_m$ is the feature dimension of each polyline point. Due to the inherent multimodality of future behaviors, the motion predictor is required to generate $K$ possible future trajectories, despite only a single ground-truth trajectory being available for supervision.

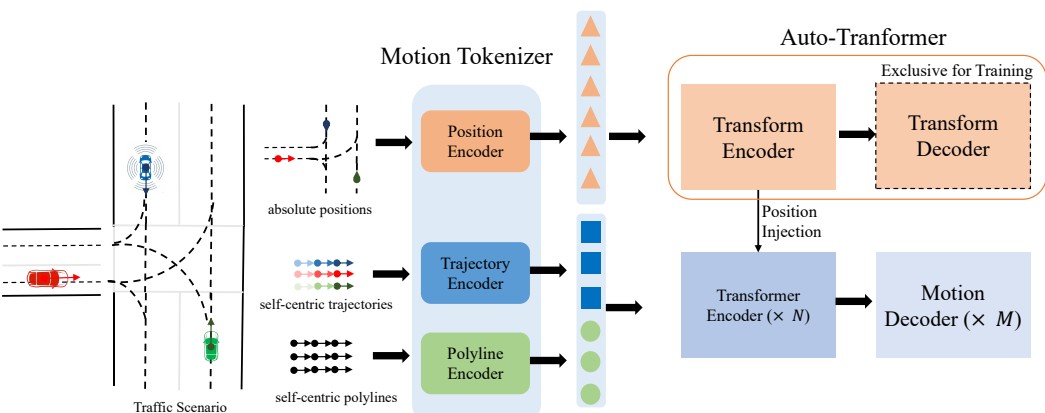

Figure 2: Illustration of a transformer-based motion predictor framework enhanced by our Trans-Abs. First, agent trajectories and map polylines are transformed into their self-centric representations, while preserving current absolute positions to maintain spatial information. Subsequently, the motion tokenizer generates agent tokens, map tokens, and absolute positional embeddings (APE). The key innovation lies in our proposed Auto-Transformer, which tames the APE to incorporate relative positional information by conduct a self-supervsied learning. The transformer encoder, integrated with the tamed APE, models spatial interactions without requiring re-encoding. Finally, the motion decoder processes the interaction features to predict future trajectories.

## 3.2 MOTION TOKENIZER

To insist on efficiency and performance, we employ a spatial-temporal factorized strategy Nayakanti et al. (2022); Shi et al. (2022) in the motion tokenizer, which first transforms the trajectories and polylines from absolute coordinate into their respective self-centric representations. Subsequently, the trajectory encoder and polyline encoder process these self-centric representations to generate agent tokens and map tokens, respectively. Meanwhile, the positional encoder directly encodes the current absolute location and heading to preserve spatial positional relationship.

### 3.2.1 SELF-CENTRIC REPRESENTATION

Given the agents $\mathbf{A} \in \mathbb{R}^{N \times t \times c_a}$ and the map polylines $\mathbf{M} \in \mathbb{R}^{M \times l \times c_m}$, we tranform them into their self-centric representation as follows:

$$
\begin{aligned}
\hat{\mathbf{A}} &= \Gamma(\mathbf{A}, \boldsymbol{l}_a, \boldsymbol{\theta}_a), \\
\hat{\mathbf{M}} &= \Gamma(\mathbf{M}, \boldsymbol{l}_m, \boldsymbol{\theta}_m),
\end{aligned}
\tag{1}
$$

where $\boldsymbol{l}_a \in \mathbb{R}^{N \times 3}$ and $\boldsymbol{l}_m \in \mathbb{R}^{M \times 3}$ denote the current location ($x$, $y$ and $z$) of the agents and map polylines in an absolute coordinate system, respectively, and $\boldsymbol{\theta}_a \in \mathbb{R}^{N \times 1}$ and $\boldsymbol{\theta}_m \in \mathbb{R}^{M \times 1}$ represent their current heading angles. The operator $\Gamma$ consists two rigid transformations: first tranlating $\mathbf{A}$ and $\mathbf{M}$ to originate at their current locations and then rotating them to align with their current orientations. This process yields the self-centric trajectories $\hat{\mathbf{A}}$ and self-centric polylines $\hat{\mathbf{M}}$.

### 3.2.2 TRAJECTORY AND POLYLINE ENCODER

This module aims to obtain agent tokens $\mathbf{X}_a \in \mathbb{R}^{N \times C}$ and map tokens $\mathbf{X}_m \in \mathbb{R}^{M \times C}$ based on the self-centric trajectories and self-centric polylines, as follows:

$$
\begin{aligned}
\mathbf{X}_a &= \phi_a(\hat{\mathbf{A}}), \\
\mathbf{X}_m &= \phi_m(\hat{\mathbf{M}}),
\end{aligned}
\tag{2}
$$

where $\phi_a$ and $\phi_m$ are sequence-related networks Dey & Salem (2017); Qi et al. (2017) designed to compress the temporal dimension of trajectories and the spatial dimension of polylines, respectively.

### 3.2.3 POSITIONAL ENCODER

This module generates the absolute positional embeddings $\mathbf{P}_a \in \mathbb{R}^{N \times C}$ and $\mathbf{P}_m \in \mathbb{R}^{M \times C}$ based on the current locations and heading angles. Specifically, we first concatenate the locations and heading angles, and then employ an non-parametric position encoding, *i.e.*, sinusoidal position embedding Vaswani et al. (2017), to encode them into a unified space, as formulated below:

$$\begin{aligned} \mathbf{P}_a &= \text{PE}([\boldsymbol{l}_a, \boldsymbol{\theta}_a]), \\ \mathbf{P}_m &= \text{PE}([\boldsymbol{l}_m, \boldsymbol{\theta}_m]), \end{aligned} \tag{3}$$

where PE denotes a 4D sinusoidal positional encoder Liu et al. (2022) that generates absolute positional embeddings, where each positional element is encoded into a vector with dimension $C/4$.

## 3.3 TRANSABS

In this section, we introduce our core TransAbs, which contains an auto-transformer incorporated into a stacked transformer encoder as illustrated in Figure 2. Inspired by the relative position generation, the auto-transformer constructs a self-supervsied learning to tame the absolute position achieving relative effects based on the Hadamard representation. The stacked transformer injects the tamed absolute position into the attention calculation by a post-multiplication manner to achieve spatial interaction and relative position encoding simultaneously, eliminating the re-encoding process.

### 3.3.1 AUTO-TRANSFORMER

Motivated by the identification that relative position generation is similar with the self-attention calculation, the auto-transformer expects to unify the relative position generation and self-attention calculation into the vector dot procut. Specifically, it contains a transform encoder and a transform decoder as shown in Figure 3.(A). Inherit the terminology in standard attention mechanism, the transform encoder employ two projections on absolute position embeddings to model query position (*i.e.*, target position) and corresponding key positions, as follows:

$$\begin{aligned} \mathbf{Q}_p &= \phi_q(\mathbf{P}), \\ \mathbf{K}_p &= \phi_k(\mathbf{P}), \end{aligned} \tag{4}$$

where $\mathbf{P} \in \mathbb{R}^{(N+M) \times C}$ is the concatenation between $\mathbf{P}_a$ and $\mathbf{P}_m$. $\phi_q$ and $\phi_k$ denotes the query and key projection, respectively. $\mathbf{Q}_p \in \mathbb{R}^{(N+M) \times C}$ and $\mathbf{K}_p \in \mathbb{R}^{(N+M) \times C}$ denotes the query positions and key positions, respectively.

The generated query and key positions are incorporated into the self-attention calculation to expect modeling spatial interaction and relative position encoding, simultaneously. Therefore, the auto-transformer constructs a self-supervsied learning to ensure the Hadamard representation of query-key position pair bringing relative features. To this end, we first generate the Hadamard representatin of any query-key pairs as follows:

$$\mathbf{H}_{ij} = \mathbf{q}_i^p \circ \mathbf{k}_j^p, \tag{5}$$

where $\mathbf{q}_i^p \in \mathbb{R}^C$ and $\mathbf{k}_j^p \in \mathbb{R}^C$ are $i$-th query position and $j$-th key position, respectively. $\mathbf{H}_{ij} \in \mathbb{R}^C$ is a vector to represent the position of $j$ relative to $i$.

Subsequently, a transform decoder receives the Hadamard representation to decode relative positions with a self-supervised learning. The process is formulated as:

$$R_{ij} = \phi_d(\mathbf{H}_{ij}), \tag{6}$$

where $R_{ij} \in \mathbb{R}^4$ is the constructed relative position of $j$-th agent centered at the $i$-th target position. Thank to the auto-transformer, the Hadamard representation is able to bring the relative positional information. In the next section, we integrate the Hadamard representation into the self-attention calculation of a transformer encoder to acheive efficient spatial interaction.

### 3.3.2 TRANSABS INTEGRATION IN TRANSFORMER ENCODER

The integration of our TransAbs lies in to make a consistent between the calculation of Hadamard representation and the calculation of self-attention mechanism as shown in Figure 3.(B). We first

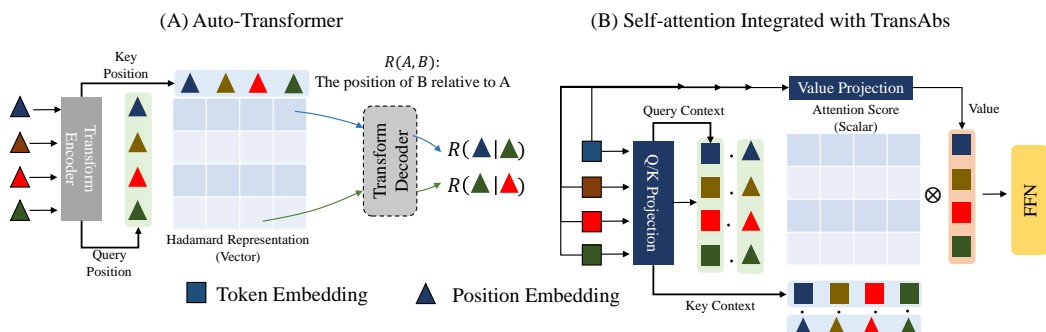

Figure 3: The Illustration of integration between our Auto-Transformer and Self-attention.

review the process of self-attention. To distinguish the auto-transformer, we call the query, key and value in self-attention mechanism as query context, key context and value context, respectively. The process of self-attention is as follows:

$$\mathbf{Q}_c = \boldsymbol{\varphi}_q(\mathbf{X}), \mathbf{K}_c = \boldsymbol{\varphi}_k(\mathbf{X}),$$
$$\mathbf{V}_c = \boldsymbol{\varphi}_v(\mathbf{X}), S = \mathbf{Q}_c \cdot \mathbf{K}_c^{\mathrm{T}}, \tag{7}$$
$$\hat{\mathbf{X}} = \frac{\mathbf{S}_c \mathbf{V}_c}{\sqrt{C}}$$

where $\mathbf{X} \in \mathbb{R}^{(N+M) \times C}$ is the input tokens obtained from the concatenation between agent tokens $\mathbf{X}_a$ and map tokens $\mathbf{X}_m$. $\boldsymbol{\varphi}_q$, $\boldsymbol{\varphi}_q$ and $\boldsymbol{\varphi}_q$ are three projections to obtain query context $\mathbf{Q}_c \in \mathbb{R}^{(N+M) \times C}$, key contex $\mathbf{K}_c \in \mathbb{R}^{(N+M) \times C}$ and value context $\mathbf{V}_c \in \mathbb{R}^{(N+M) \times C}$, respectively. $S \in \mathbf{R}^{(N+M) \times (N+M)}$ is the atttention scores matrix. $\sqrt{C}$ is a scaling factor to ensure numerical stability. $\hat{\mathbf{X}}$ is the output tokens.

To incorporate the Hadamard representation into this self-attention process, we split the calculation of attention scores $S$ into two steps: the first is Hadamard product and the second is vector sum as:

$$S_{ij} = \sum_{0}^{C-1} \mathbf{q}_i^c \circ \mathbf{k}_j^c, \tag{8}$$

where $\mathbf{q}_i^c \in \mathbb{R}^C$ is the $i$-th query context and $\mathbf{k}_j^c \in \mathbb{R}^C$ is the $j$-th key context. $S_{ij} \in \mathbb{R}^1$ is an attention score. This formulation is fully consitent with the process of Hadamard representation, we incorporate them as follows:

$$\hat{S}_{ij} = \sum_{0}^{C-1} (\mathbf{q}_i^c \circ \mathbf{q}_i^p) \circ (\mathbf{k}_j^c \circ \mathbf{k}_j^p), \tag{9}$$

where $\hat{S}_{ij} \in \mathbb{R}^1$ is an attention score enhanced by our Hadamard representation. Since multiplication satisfies the associative law, the Equation 9 can be transformed into:

$$\hat{S}_{ij} = \sum_{0}^{C-1} (\mathbf{q}_i^c \circ \mathbf{k}_j^c) \circ (\mathbf{q}_i^p \circ \mathbf{k}_j^p), \tag{10}$$

where the second item $\mathbf{k}_j^c \circ \mathbf{k}_j^p$ provides the relative position information based on our auto-transformer, the first item $\mathbf{q}_i^c \circ \mathbf{k}_j^c$ provides the motion contexts, thus modeling the spatial interaction and relative position encoding simultaneously.

To leverage the parallel computation of GPUs, the Equation 9 is tranformed back to its matrix forms:

$$\hat{S} = (\mathbf{Q}_c \circ \mathbf{Q}_p) \cdot (\mathbf{K}_c \circ \mathbf{K}_p)^{\mathrm{T}}, \tag{11}$$

where $\hat{S} \in \mathbb{R}^{(N+M) \times (N+M)}$ is used to replace the $S$ in Equation 7 to acheive efficient spatial interaction.

For the multi-head self-attention with $h$ heads, we divide the Hadamard representation into $h$ sub-vectors with $C//h$ dimensions. We use $h$ transform decoders to decode the sub-vectors into corresponding relative positions.

### 3.3.3 MODEL TRAINING

Our TransAbs is trained jointly with the motion predictor to align the Hadamard representation (*i.e.*, the second item in Equation 10) and the context attention score (*i.e.*, the first item in Equation 10). Therefore, there are two types of loss function: self-supervised loss and trajectory prediction loss.

The self-supervised loss comes from our auto-transformer to decode relative position based on a Hadamard representation. However, there could occur large memory consumption in the generation of Hadamard representation as shown in Equation 5, such as $N + M$ is greater than 1000 in a autonomous traffic scenario. To address this issue, we employ a **random decoding** strategy in auto-transformer. Specifically, we random select $n << N$ agents and $m << M$ map polylines to generate Hadamard representation and decode them to their relative positions. Note that we feed all query positions and key positions to the transformer encoder due to the efficient matrix multiplication in self-attention. The self-supervised loss is formulated as :

$$\mathcal{L}_{ssl} = ||R, \hat{R}||, \tag{12}$$

where $|| \cdot ||$ denotes a regression loss. $R \in \mathbb{R}^{(n+m)\times(n+m)\times 5}$ is the predicted relative positions and $\hat{R} \in \mathbb{R}^{(n+m)\times(n+m)\times 5}$ is the ground-truth. Note that we take the sinusoidal value and cosine value of the heading angle to avoid the periodic influences.

The trajectory prediction loss is consistent with the baseline method. It commonly has a regression loss and classification loss as follows:

$$\mathcal{L}_{tp} = \lambda_1 ||Y, \hat{Y}|| + \lambda_2 \text{CE}(i, P), \tag{13}$$

where $Y \in \mathbb{R}^{N\times T\times c_a}$ is the best predicted trajectory with index $i$ selected from the predicted diverse trajectories $\mathbf{Y} \in \mathbb{R}^{N\times K\times T\times c_a}$. $P \in \mathbb{R}^{N\times K}$ is the confidence of each predicted trajectories. CE is the cross-entropy loss. $\lambda_1$ and $\lambda_2$ are loss weights to balance two losses. The whole model is trained with an end-to-end manner as follows:

$$\mathcal{L} = \mathcal{L}_{ssl} + \mathcal{L}_{tp}. \tag{14}$$

### 3.3.4 MODEL INFERENCE

Our auto-transformer tames the Hadamard representation to bring relative positional information with a self-supervsied learning. In model inference stage, we discard the generation of Hadamard representation and transform decoder similar with the autoencoders. Namely, we diretly transport the query position and key position to the transformer encoder to achieve relative interaction without any re-encoding process.

## 4 EXPERIMENTS

### 4.1 DATASETS

We evaluate the effectiveness of our TransAbs on a large scale autonomous driving benchmark, Waymo Open Motion Dataset (WOMD) Ettinger et al. (2021). WOMD contains $487k$ training scenes, $44k$ validation scenes and $44k$ testing scenes. It provides 1 second history data, including agents (vehicle, pedestrian, cyclist) and map polylines. The model predicts the next 8 seconds future trajectory. Note that all experiments are conducted on the latest dataset version, *i.e.* WOMD v1.3.0. We use the official metrics, mAP, to evaluate our proposed method.

### 4.2 IMPLEMENTATION DETAILS

We integrate our TransAbs into two strong baseline methods: MTR++ Shi et al. (2024) and MTR Shi et al. (2022). MTR employs full relative interaction to model spatial relationships for each target agent separately, while MTR++ is an enhanced version that uses present-only relative interaction for spatial modeling. For MTR integration, we utilize its official open-source codebase with default experimental configurations. Since MTR++ is a closed-source model, we reproduce it based on MTR with the default settings described in the original paper. For TransAbs implementation, the

query and key projections in the auto-transformer are implemented using an MLP with layer normalization, respectively. The transform decoder consists 8 learnable linear layers to decode relative positions in different attention spaces. To avoid out-of-memory (OOM) issues, we randomly select 100 agents and map polylines to randomly decode them into their relative positions.

| Method | Agent Category | mAP ↑ | minADE ↓ | minFDE ↓ | Miss Rate ↓ |
|---|---|---|---|---|---|
| 1) MTR++ Shi et al. (2024) | VEHICLE | 0.4597 | 0.7541 | 1.5083 | 0.1509 |
| | PEDESTRIAN | 0.4770 | 0.3539 | 0.7349 | 0.0762 |
| | CYCLIST | 0.3543 | 0.7181 | 1.4362 | 0.1864 |
| | Avg | 0.4303 | 0.6087 | 1.2265 | 0.1378 |
| 2) APE + MTR++ | VEHICLE | 0.4426 | 0.7751 | 1.5579 | 0.1591 |
| | PEDESTRIAN | 0.4703 | 0.3591 | 0.7495 | 0.0780 |
| | CYCLIST | 0.3526 | 0.7446 | 1.5033 | 0.1934 |
| | Avg | 0.4218 | 0.6263 | 1.2702 | 0.1435 |
| 3) TransAbs + MTR++ | VEHICLE | 0.4585 | 0.7605 | 1.5264 | 0.1523 |
| | PEDESTRIAN | 0.4794 | 0.3523 | 0.7340 | 0.0772 |
| | CYCLIST | 0.3758 | 0.7231 | 1.4495 | 0.1848 |
| | Avg | **0.4379** | 0.6120 | 1.2366 | 0.1381 |

Table 1: Performance comparisons on the validation set of Waymo Open Motion Dataset. mAP is the primary metric. Bold values indicate the best average performance across all categories, while underlined values represent the best performance for specific agent categories.

| | MTR | MTR+APE | MTR+RPE | MTR+RPB | MTR+RoPE | MTR+TransAbs |
|---|---|---|---|---|---|---|
| mAP | 0.4345 | 0.4294 | 0.4332 | 0.4329 | 0.4303 | **0.4357** |

Table 2: TransAbs verus different positional embeddings on the validation set of Waymo Open Motion Dataset.

### 4.3 TransAbs verus Present-only Relative Interaction

Table 1 presents performance comparisons on the WOMD validation set between our TransAbs and the present-only relative interaction model, MTR++ Shi et al. (2024). We evaluate the effectiveness of TransAbs through three MTR++ variants: 1) the baseline MTR++ with default settings; 2) MTR++ enhanced with absolute positional embeddings (APE); and 3) MTR++ integrated with our TransAbs. The experimental results demonstrate that relative interaction outperforms absolute interaction, while our TransAbs effectively transforms absolute interactions into relative effects, achieving the best mAP performance. Across different agent categories, TransAbs exhibits more balanced performance, achieving the best mAP for pedestrians and cyclists while maintaining comparable performance to MTR++ for vehicles. Regarding secondary metrics, our method shows minimal performance differences compared to MTR++, while significantly outperforming the APE+MTR++.

### 4.4 TransAbs verus Different Positional Embeddings

We further validate the effectiveness of our TransAbs against different positional embeddings, including absolute positional embedding (APE), relative position embedding (RPE) Shaw et al. (2018), relative positional bias (RPB) Liu et al. (2021b) and rotation positional embedding (RoPE) Zhao et al. (2025). As shown in Table 2, we use the MTR Shi et al. (2022) as the baseline and our TransAbs showcases the best mAP performance. In addtion, we find that other positional embeddings reduce the baseline performances.

### 4.5 Computational Efficiency

TransAbs efficiently generates relative positional information by taming absolute interactions without requiring any re-encoding process. We conduct comprehensive comparisons across absolute

positional embeddings (APE+MTR++), present-only relative positional embeddings (MTR++) and full relative positional embeddings (MTR). As demonstrated in Table 3, our TransAbs achieves superior performance in both computational efficiency (reduced GPU memory usage and inference latency) and prediction accuracy (improved mAP) while maintaining minimal parameter overhead compared to existing approaches.

| Method | Parameters | Memory | Latency |
|---|---|---|---|
| APE+MTR++ | 69.7 M | 557.41 M | 63.44 ms |
| MTR++ | 71.3 M | 682.32 M | 73.25 ms |
| TransAbs+MTR++ | 71.7 M | 566.97 M | 67.46 ms |
| MTR | 65.7M | 538.34 M | 48.95 ms |
| TransAbs+MTR | 65.2 M | 300.09 M | 48.30 ms |

Table 3: Efficiency comparisons of different methods.

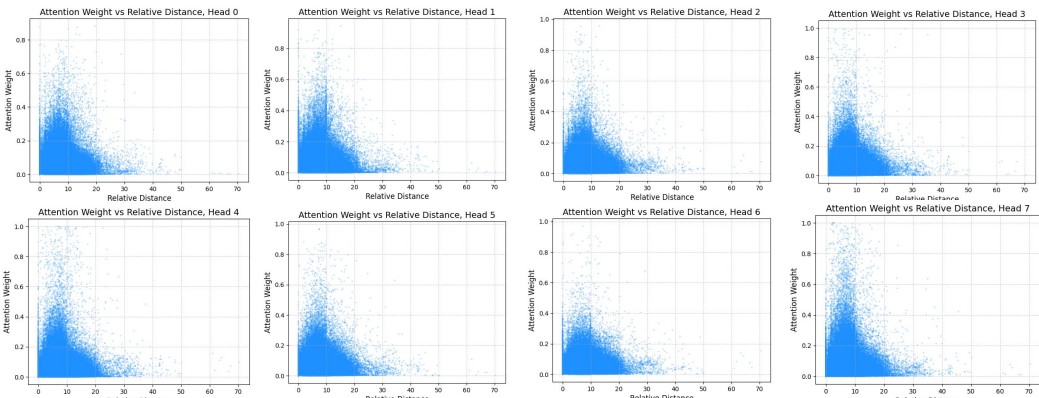

Figure 4: Visualizations of the attention weights on the Waymo Open Motion Dataset validation set.

### 4.6 VISUALIZATION OF ATTENTION WEIGHTS ENHANCED BY OUR TRANSABS

As illustrated in Figure 4, we visualize the attention scores enhanced by our TransAbs. The figure clearly demonstrates that higher attention scores are predominantly concentrated within the range of $[0, 20]$. Two distinct patterns emerge: first, a long-term decay, indicating that distant traffic participants exert less influence; and second, a moderate-distance dependence, which may be attributed to the influence of future waypoint planning.

## 5 CONCLUSIONS

In this paper, we present TransAbs, a novel approach that addresses the computational inefficiency of relative interaction modeling in motion prediction while maintaining high prediction accuracy. Our key insight is that the relative position generation process can be unified with self-attention calculation through Hadamard representation, enabling efficient spatial interaction modeling without the need for redundant re-encoding. Extensive experiments on the Waymo Open Motion Dataset demonstrate that TransAbs successfully balances prediction accuracy and computational efficiency, achieving comparable or superior performance to state-of-the-art relative interaction approaches while significantly reducing computational overhead.

The implications of this work extend beyond motion prediction, as the principle of transforming absolute interactions to achieve relative effects while maintaining computational efficiency could be applied to other domains requiring spatial relationship modeling, offering a practical solution to advance the field toward efficient and scalable autonomous driving systems.

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

## A  APPENDIX

### A.1  EXTRA ABLATION STUDIES

#### A.1.1  DIFFERENT ABSOLUTE POSITIONAL EMBEDDINGS

Our TransAbs transforms absolute positional embeddings to generate relative positional information. To identify the optimal absolute positional embedding (APE) variant, we conduct an ablation study comparing three different approaches. As shown in Table 4, we evaluate: 1) direct input of naive absolute positions into the transform encoder to obtain query and key positions; 2) encoding of 2D location and heading using sinusoidal positional encoding; and 3) encoding of 3D location and heading using sinusoidal positional encoding. The experimental results demonstrate that sinusoidal positional encoding significantly improves APE performance, with 3D encoding (including the $z$-axis) outperforming 2D encoding (using only $x$ and $y$ axes).

#### A.1.2  EFFECTS OF MULTIPLICATION POSITIONAL EMBEDDING

Our TransAbs employs multiplication positional embedding between query/key context and query/key position to inject relative positional information. To evaluate the effectiveness of this approach, we conduct an ablation study comparing three variants. As shown in Table 5, we evaluate: 1) addition positional embedding between query/key context and query/key position; 2) multiplication embedding with gradient stopping from query/key position to query/key context; and 3) our full TransAbs with end-to-end optimization. The experimental results demonstrate that multiplication positional embedding significantly outperforms addition-based approaches, and joint optimization between TransAbs and the motion predictor yields superior performance compared to individual optimization strategies.

| Variants | minADE ↓ | minFDE ↓ | mAP ↑ |
|---|---|---|---|
| 1) | 0.6110 | 1.2356 | 0.4267 |
| 2) | 0.6168 | 1.2479 | 0.4327 |
| 3) | 0.6120 | 1.2366 | **0.4379** |

Table 4: Ablation studies across differnet absolute positional embeddings.

| Variants | minADE ↓ | minFDE ↓ | mAP ↑ |
|---|---|---|---|
| 1) | 0.6177 | 1.2494 | 0.4236 |
| 2) | 0.6181 | 1.2478 | 0.4255 |
| 3) | 0.6120 | 1.2366 | **0.4379** |

Table 5: Ablation studies across differnet positional embeddings.

#### A.1.3  VISUALIZATION OF THE PREDICTED FUTURE TRAJECTORIES

We provide additional visualizations of the predicted multimodal future trajectories in Figure 5. Each target agent predicts six possible future trajectories. Notably, our model is capable of predicting the trajectories for all target agents in a single forward pass. Our method demonstrates robust performance across diverse motion scenarios, generating accurate and reliable predictions. Specifically, subfigures (1), (2), and (3) show predictions for different agents exhibiting various future behaviors (turning left/right, going straight, or remaining stationary); (4) illustrates the prediction for two interacting vehicles; (5) demonstrates predictions in a denser scenario; and (6) depicts a long-tail behavior, *i.e.*, U-turn.

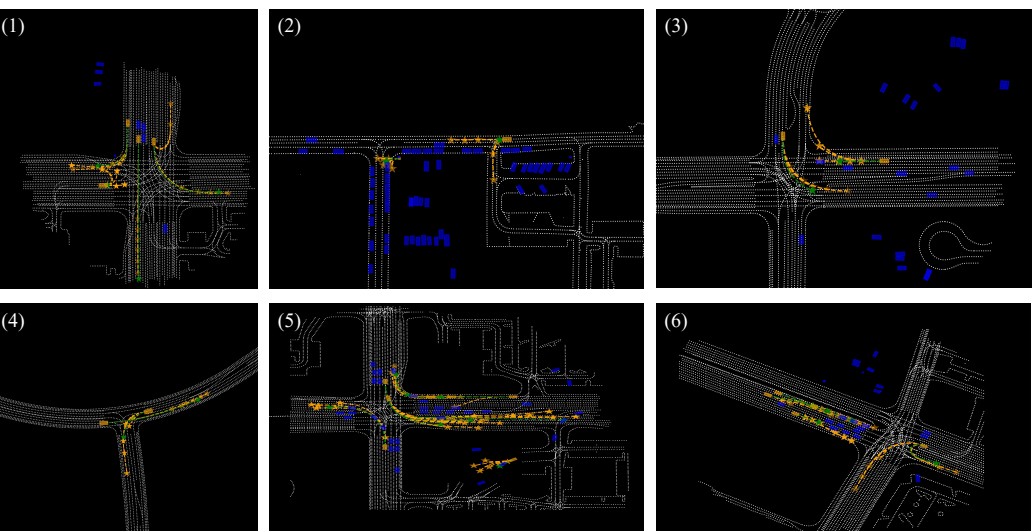

Figure 5: Visualization of predicted multimodal future trajectories on the Waymo Open Motion Dataset validation set. White lines represent map polylines, the orange bounding boxes indicates the target agents, blue bounding boxes show neighboring agents, the green lines are the ground-truth trajectories, and orange dashed lines depict the predicted future trajectories.

