# OpenReview forum: "TransAbs: Taming Absolute Interaction for Efficient Relative Motion Prediction"
_ICLR.cc/2026/Conference — Submitted to ICLR 2026_

### Official Review · Reviewer_hsDV · 2025-10-29

**Soundness:** 2
**Presentation:** 3
**Contribution:** 2
**Rating:** 4
**Confidence:** 4

**Summary:**

The paper proposes TransAbs, a module that performs relative position generation into the self-attention process for efficient multi-agent motion prediction in autonomous driving. Instead of repeatedly re-encoding relative coordinates for each agent, TransAbs introduces a Hadamard-based Auto-Transformer that learns to convert absolute positional embeddings into relative ones through a self-supervised alignment mechanism. By injecting these tamed embeddings into the attention computation through post-multiplicative encoding, the model achieves relative spatial reasoning with lite computations. Integrated into transformer predictors such as MTR orMTR++, TransAbs delivers comparable or slightly improved accuracy on the Waymo Open Motion Dataset while reducing memory and latency.

**Strengths:**

1. Reformulating relative transformation as an implicit Hadamard alignment between absolute position embeddings and attention computation is conceptually appealing and potentially generalizable;

2. The insights to tackle relative interactions inside attention seems appealing.

**Weaknesses:**

1. The proposed transformation is described heuristically, without formal analysis of when or why the Hadamard mapping preserves relative geometric consistency.

2. Testing result of MTR+TransAbs on WOMD Leaderboard is missing. Recent efficient or unified-interaction frameworks such as Wayformer, ModeSeq, or QCNet are not quantitatively evaluated.

3. It’s unclear how TransAbs scales to larger agent counts.

4. More explanation on stability, convergence, and whether this loss is weighted or balanced against the main prediction loss would improve clarity

5. Qualitative analysis is pretty limited. In depth analysis of the mechanism should also be added

**Questions:**

1. How does TransAbs behave in extremely dense urban scenes?does memory scale linearly or quadratically after pruning?

2. How sensitive is performance to the number of sampled pairs in terms of self-supervised learning objective?

3. Have you tested TransAbs on other benchmarks like Argoverse2 or nuScenes to confirm dataset generalization?

4. How is the performance combining TransAbs on other predictors / planners such as QCNet or DeMo?

---

### Official Review · Reviewer_9dxQ · 2025-11-02

**Soundness:** 2
**Presentation:** 2
**Contribution:** 2
**Rating:** 4
**Confidence:** 4

**Summary:**

This paper proposes TransAbs, a transformer-based module designed to make absolute spatial interactions behave like relative ones without explicit re-encoding. The key idea is to incorporate relative positional effects directly into the attention mechanism through a Hadamard representation between pairs of absolute positions. By aligning this representation with attention scores through a self-supervised auto-transformer, the method aims to retain the efficiency of absolute interaction while capturing the precision of relative modeling.
Experiments on the Waymo Open Motion Dataset (WOMD) show that TransAbs, when integrated into MTR++, achieves slightly higher accuracy.

**Strengths:**

1.	The paper provides a clear motivation: relative interaction methods yield higher accuracy but are computationally expensive, and TransAbs attempts to combine the benefits of both paradigms.

2.	The integration of a self-supervised auto-transformer into the attention mechanism is theoretically sound.

3.	The presentation of the Hadamard-based formulation is clear, and the efficiency analysis is helpful for understanding potential runtime benefits.

**Weaknesses:**

1.	Insufficient experimental validation.

The evaluation relies almost entirely on MTR++ as the backbone, which is not the best-performing model on WOMD and is not open-sourced. Using only this backbone is restrictive and does not prove the generality of TransAbs. Integrating the method into stronger backbones or into a variety of architectures would be essential for validating the claim of broad applicability.

2.	Limited comparative scope.

Tables 1 and 2 compare TransAbs only with a few positional encodings (APE, RPE, RPB, RoPE). In Section 2.0.2, the paper discusses other relative-encoding frameworks such as HiVT, QCNet, and Simpl, but no experiments or quantitative results against these approaches are presented. Without these comparisons, it is unclear whether TransAbs offers a meaningful accuracy improvement over established relative-encoding models.

3.	Weak empirical gains.

The mAP improvements shown in Tables 1–2 are very small, often within the margin of training variance. The claim that TransAbs “balances prediction accuracy and computational efficiency” is not strongly supported by the reported numbers. A more comprehensive set of ablations and statistical confidence intervals would be required to establish significance.

4.	Overstated claims of generality.
The method is presented as a general way to replace relative interaction, but all results are restricted to one dataset and one baseline. There is no evidence that the technique would transfer to other prediction settings or benchmarks.

**Questions:**

Why was MTR++ selected as the sole backbone? Did you attempt integration with stronger or more recent models?

Would the proposed Hadamard-based positional encoding still be effective when applied to non-transformer motion predictors?

---

### Official Review · Reviewer_3WY1 · 2025-11-04

**Soundness:** 2
**Presentation:** 2
**Contribution:** 2
**Rating:** 2
**Confidence:** 3

**Summary:**

This paper proposes TransAbs, a transformer-based motion prediction framework that tackles the efficiency-accuracy trade-off in spatial interaction modeling. The approach addresses two pain points: (1) an Auto-Transformer module that learns to map Hadamard representations of absolute position pairs to relative positions via self-supervised learning, avoiding the expensive coordinate re-encoding required by traditional relative methods; and (2) a post-multiplication positional encoding strategy that integrates the learned relative information directly into the attention mechanism, enabling simultaneous spatial interaction and relative position encoding in a single forward pass. The model is trained end-to-end with both self-supervised loss and trajectory prediction loss. Experiments on Waymo Open Motion Dataset show that when integrated into MTR and MTR++ baselines, TransAbs achieves comparable or marginally improved accuracy (~1-2% mAP gain) with reduced memory usage and inference latency.

**Strengths:**

- TransAbs explicitly addresses the re-encoding overhead of relative interaction methods through its Auto-Transformer module, which learns to encode relative positions from Hadamard representations of absolute positions. The post-multiplication integration directly eliminates redundant coordinate transformations while maintaining relative positional effects—addressing a known computational bottleneck in motion prediction.
- TransAbs is positioned as a single approach that (i) operates on absolute coordinates for efficiency, (ii) achieves relative positional modeling through learned representations, and (iii) integrates seamlessly into existing transformer architectures.
- TransAbs employs a training-inference decoupling design. The transform decoder and Hadamard representation generation are used only during training for self-supervised learning, then discarded at inference.

**Weaknesses:**

- Insufficient theoretical justification for the core claim. The paper asserts that the Hadamard product encodes relative position information (Eq. 9-10), but this is not rigorously justified. Element-wise multiplication followed by summation does not obviously capture the geometric transformations (translation + rotation) that define relative positions. The Auto-Transformer learns a mapping via self-supervised loss, but there's no analysis of (i) what representations it actually learns, (ii) why Hadamard products specifically enable this.
- Marginal performance gains with trade-offs in other metrics. While TransAbs shows modest mAP improvements (+1.8% on MTR++: 0.4379 vs 0.4303), Table 1 reveals that other important metrics actually degrade or show mixed results. For MTR++, minADE increases from 0.6087 to 0.6120, minFDE increases from 1.2265 to 1.2366, indicating worse prediction accuracy in terms of displacement errors.
- Questionable efficiency claims. Table 3 shows a paradox: TransAbs+MTR++ has more parameters yet uses less memory and has lower latency. This counterintuitive result is not explained. Without clarification, it's unclear whether the efficiency gains are genuine or artifacts of implementation choices.
- Inconsistent baseline comparisons and reproducibility concerns. Table 3 shows TransAbs+MTR has fewer parameters (65.2M) than MTR alone (65.7M), which is counterintuitive given that TransAbs adds components. This suggests either an error or undisclosed architectural changes. Additionally, critical hyperparameters are missing: loss weights ($\lambda_1$, $\lambda_2$ in Eq. 13)​, learning rates, and training schedules are not specified, making reproduction difficult.
- Limited scope of evaluation. The method is only tested on WOMD and only integrated into two baselines (MTR/MTR++). Testing on other datasets (Argoverse 2 Motion Forecasting Dataset) would demonstrate generalizability.
- Presentation clarity issues. (i) Figure 3's "Hadamard Representation" visualization is hard to understand. Showing actual tensor shapes and data flow would help, (ii) Section 3.3.1's claim that "relative position generation is similar with self-attention" requires more rigorous comparison before introducing the Auto-Transformer;

**Questions:**

- Parameter inconsistency. Why does TransAbs+MTR (65.2M) have fewer parameters than MTR (65.7M) in Table 3? Are there architectural modifications not mentioned in the text? Please clarify what accounts for this difference.
- Hyperparameter specifications. What are the values of $\lambda_1$​, $\lambda_2$ in Eq. 13? How is the relative weighting between $L_{ssl}$ and $L_{tp}$​ chosen? How sensitive is performance to these choices? Please report the final values used.
- Generalization beyond WOMD. Can you test TransAbs on other motion prediction datasets (Argoverse 2 Motion Forecasting Dataset) or integrate it into other transformer-based architectures to demonstrate broader applicability?
- Theoretical justification and learned representations. Provide (i) a mathematical proof or deeper analysis showing why the Hadamard product encodes relative position, (ii) visualizations of the decoded relative positions vs. ground truth,  and (c) analysis of what geometric or algebraic properties this representation preserves. Providing this analysis would strengthen the theoretical foundation.

---

### Meta-Review · Area_Chair_bzZG · 2025-12-31

**Summary:**

Reviewer concerns include insufficient experimental evaluation and marginal improvements on those that were done, unjustified / excessive claims, and poor clarity of the paper.

**Reviewer Concerns:**

There were no rebuttal comments, and no apparent revisions to the paper. Hence all the reviewer concerns remain outstanding.

**Reviewer Scores:**

The original reviewer scores were 4, 4, 2, so all are already leaning to reject. Given the lack of rebuttal comments, there is no reason for the reviewer scores to improve.

---

### Decision · Program_Chairs · 2026-01-26

Reject